# Poly(amidoamine) Dendrimer/Camptothecin Complex: From Synthesis to In Vitro Cancer Cell Line Studies

**DOI:** 10.3390/molecules28062696

**Published:** 2023-03-16

**Authors:** Ewa Oledzka, Klaudia Paśnik, Izabela Domańska, Monika Zielińska-Pisklak, Urszula Piotrowska, Marcin Sobczak, Łukasz Szeleszczuk, Anna Laskowska

**Affiliations:** 1Department of Pharmaceutical Chemistry and Biomaterials, Faculty of Pharmacy, Medical University of Warsaw, Banacha 1 Str., 02-097 Warsaw, Poland; 2Military Institute of Hygiene and Epidemiology, Kozielska 4 Str., 01-163 Warsaw, Poland; 3Department of Physical Chemistry, Faculty of Pharmacy, Medical University of Warsaw, Banacha 1 Str., 02-097 Warsaw, Poland; 4Department of Pharmaceutical Microbiology, Centre for Preclinical Research, Medical University of Warsaw, Banacha 1b Str., 02-097 Warsaw, Poland

**Keywords:** PAMAM dendrimer, camptothecin, encapsulation, anti-cancer drug delivery systems, controlled release, drug carrier, non-small-cell lung cancer

## Abstract

Camptothecin (CPT), an alkaloid with potent anticancer activity, is still not used in clinical practice due to its high hydrophobicity, toxicity, and poor active-form stability. To address these shortcomings, our research focuses on the encapsulation of this drug in the poly(amidoamine) (PAMAM) dendrimer macromolecule. The PAMAM dendrimer/CPT complex was synthesized and thoroughly characterized. The in vitro drug release study revealed that the drug was released in a slow and controlled manner in acidic and physiological conditions and that more than 80% of the drug was released after 168 h of incubation. Furthermore, it was demonstrated that CPT was released with first-order kinetics and non-Fickian transport. The studies on the hemolytic activity of the synthesized complex indicated that it is hemocompatible for potential intravenous administration at a concentration ≤ 5 µg/mL. Additionally, the developed product was shown to reduce the viability of non-small-cell lung cancer cells (A549) in a concentration- and time-dependent manner, and cancer cells were more susceptible to the complex than normal fibroblasts. Lastly, molecular modeling studies revealed that the lactone or carboxylic forms of CPT had a significant impact on the shape and stability of the complex and that its formation with the lactone form of CPT was more energetically favorable for each subsequent molecule than the carboxylic form. The report represents a systematic and structured approach to develop a PAMAM dendrimer/CPT complex that can be used as an effective drug delivery system (DDS) for the potential treatment of non-small-cell lung cancer.

## 1. Introduction

Despite significant advancements in conventional treatment methods such as chemotherapy and radiation, cancer therapy remains far from ideal due to a number of limitations. Current cancer therapies frequently face challenges such as non-specific systemic antitumor agent distribution, insufficient drug concentrations reaching the tumor site, excessive cytotoxicity, limited capacity to evaluate therapeutic responses, and the development of multiple-drug resistance [1,2,3]. Existing diagnostic and prognostic categorization is inadequate to predict effective therapy and patient outcome [4]. As a result, there is an urgent need and significant opportunity to develop new and creative technologies that can assist to delineate tumor margins, detect remaining tumor cells and micrometastases, and assess if a tumor has been entirely eliminated.

Camptothecin (CPT) is an inhibitor of the DNA-replicating enzyme topoisomerase I, which is believed to act by stabilizing a topoisomerase I-induced single-strand break in the phosphodiester backbone of DNA, thereby preventing relegation [5]. This causes a double-strand DNA break during replication, which leads to cell death if not repaired [6]. CPT, a naturally occurring alkaloid, was identified from the tree *Camptotheca acuminata* in China in 1966 [7]. CPT has been proven in preclinical tests to be effective against human xenografts of colon, lung, breast, ovarian, and melanoma cancers [8,9]. Despite the promise shown in preclinical studies, clinical trials were stopped due to unforeseen toxicity and poor antineoplastic efficacy [5]. Furthermore, CPT was considered to have limited therapeutic potential due to its poor solubility, and it has been postulated that local administration of CPT might be a way to acquire effective drug concentrations in brain tumors without the unwanted side effects associated with systemic distribution [5].

PAMAM dendrimers are a class of synthetic macromolecules that are highly branching and monodisperse, with well-defined structures and compositions. These nanocarriers can be synthesized using a divergent approach [10]. Internal cavities and peripheral functional groups (e.g., amine, hydroxyl, ester, etc.) of PAMAM dendrimers can be altered to encapsulate drugs or other active substances. Using this nanomaterial, interactions between PAMAM and drugs may be controlled. Non-immunogenicity, water solubility, spherical shape, acceptable biodegradation, biocompatibility, minimal nonspecific blood-protein binding, and controlled drug release are further features of these nanomaterials that make them appropriate carriers to deliver drugs and genes. They have also been widely employed as co-delivery systems for the simultaneous administration of genes and drugs [11,12,13]. Hydrophobic drugs with different mechanisms of action and low solubility can be physically encapsulated or entrapped inside PAMAM dendrimers’ cavities or pockets [14,15,16,17,18,19]. This molecular encapsulation is responsible for the increased solubility in water and controlled release profile. It is essential to note that the physical interactions between PAMAM and drugs in complexes in aqueous solutions can be regulated by a variety of noncovalent interactions such as hydrogen bonding, electrostatic interactions, steric hindrance, and hydrophobic interactions [13]. These interactions may have an impact on the drug release profile. The amine groups remain deprotonated during the drug release process, while the branches come together by compressing into the central core. This mechanism regulates the drug release processes in different conditions. For example, because the microenvironment of the tumor vasculature is acidic, the amine groups will be protonated, and their conformation will be changed. A low pH can offer an adequate stimulus for drug release. As a result, drug release from PAMAM dendrimers is always pH-sensitive and occurs faster in acidic environments [20,21].

Until now, only a few extensive studies have been conducted to evaluate dendrimers, especially PAMAM dendrimers, as carriers of encapsulated CPT and its synthetic derivatives. Namely, Morgan and coworkers [22] used a biocompatible polyester, not a PAMAM dendrimer, composed of the natural metabolites glycerol and succinic acid (PGLSA) for the encapsulation of 10-hydroxycamptothecin (10-HCPT) and 7-butyl-10-aminocamptothecin (BACPT). The authors investigated the cytotoxicity of these supramolecular assemblies toward human breast adenocarcinoma (MCF-7), colorectal adenocarcinoma (HT-29), non-small-cell lung carcinoma (NCI-H460), and glioblastoma (SF-268) cell lines as well as the effect of encapsulating 10-HCPT within PGLSA dendrimers on drug uptake and efflux rates when exposed to MCF-7 cells. They found that cellular uptake and efflux measurements in MCF-7 cells show a 16-fold increase for cellular uptake and an increase in drug retention within the cell when using the dendrimer vehicle [22]. Some other study indicates that PAMAM dendrimers could be used to improve the delivery of 7-ethyl-10-hydroxy-camptothecin (SN-38), a potent topoisomerase I poison and a biologically active metabolite of irinotecan hydrochloride (CPT-11) [23]. The authors synthesized complexes of SN-38 with generation 4 (G4.0) amine-terminated PAMAM dendrimers with varying amounts of the drug to assess complex stability as well as the effect of complexation on permeability across and cellular uptake by Caco-2 cells. In that study, it was discovered that the complexes were stable at pH 7.4 and that the drug was released at pH 5.0. In comparison to free SN-38, the complexes had a tenfold increase in permeability and a hundredfold increase in cellular uptake. However, further work is required to prepare SN-38 dendrimers containing PAMAM that will be stable in the harsh gastric environment. Interesting work was presented by Sadekar and coworkers [24], where CPT was formulated and co-delivered with cationic, also amine-terminated PAMAM dendrimer (G4.0) and anionic, carboxylate-terminated PAMAM dendrimer (G3.5). The authors discovered that both PAMAM dendrimers G4.0 and G3.5 increased drug solubilization in simulated gastric fluid and caused a 2–3-fold increase in the oral absorption of CPT when delivered at 2 h. Furthermore, G4.0 and G3.5 PAMAM dendrimers did not increase mannitol transport, indicating that the oral absorption of CPT was not due to tight junction modulation, and histologic observations of the epithelial layer of small intestinal segments of the gastrointestinal tract (GIT) at 4 h post dosing supported no evidence of toxicity at the evaluated doses of PAMAM dendrimers. In contrast, another study attempted to develop multifunctional dendrimeric support with long hydrophobic C12 alkyl chains for the simple complexation of 10-HCPT [25]. It was discovered that the complex had a high loading efficiency, with each molecule encapsulating approximately 20 drug molecules; high stability, with no detectable drug release during dialysis for three days; and high water solubility, achieving a 600-fold increase over the water solubility of free 10-HCPT. Furthermore, it exhibited significantly higher cytotoxicity against 22RV1 cells overexpressing integrin αvβ3 and significantly lower cytotoxicity against MCF-7 cells, which express low levels of integrin αvβ3. In the last study on a related subject, Yiyun Cheng and colleagues focused their laboratory work on utilizing G4.0, G.05, and G6.0 PAMAM dendrimers with an ethylenediamine (EDA) core for CPT encapsulation/complexation. The authors, however, limited their investigations to determining the effect of PAMAM generation on CPT solubility [26]. As a consequence, we decided to go further on that research and use a PAMAM dendrimer nanocarrier to encapsulate CPT in order to prolong drug release and improve its solubility in the treatment of non-small-cell lung cancer in vitro. The synthesized solid complex was thoroughly evaluated in structural and biological terms, including toxicity testing, as well as theoretically using molecular modeling methods. To the best of our knowledge, such studies have not yet been performed and documented in the literature.

## 2. Results and Discussion

### 2.1. Synthesis and Characterization of the PAMAM Dendrimer/CPT Complex

CPT has low oral bioavailability and low stability due to the rapid hydrolysis of its ɑ-hydroxylactone ring under physiological conditions [27]. Unfortunately, the closed ɑ-hydroxylactone moiety (E-ring) is an important structural requirement for CPT’s successful interaction with the topoisomerase I target and antitumor potency in vivo. Furthermore, CPT is extremely insoluble in water and in most organic solvents [28]. In an attempt to reduce these drawbacks, the development of controlled-release formulations for oral delivery, containing encapsulated CPT appears to be a promising strategy, particularly with the use of PAMAM dendrimers, where the drug can be physically encapsulated or entrapped within the internal voids or pockets of PAMAM dendrimers. Given the aforementioned, the first stage of our investigation was the synthesis and characterization of the G4.0 PAMAM dendrimer/CPT complex (Figure 1). The synthesis was carried out in a DMF solution and gives a calculated encapsulation efficiency (*EE*) value of 68.24%.

Figure 2 represents the ^1^H NMR spectra of the synthesized complex, whereas SuppInfo Materials show the spectra with the corresponding chemical shifts of a pure CPT and a G4.0 PAMAM dendrimer (Appendix A). The spectrum demonstrates relevant signals derived from both substances, PAMAM dendrimer, and CPT. When evaluating them, it is important to consider the visually evident chemical shifts of the signals a, e, f, and b in comparison to the free active substance as a result of CPT being encapsulated into the interior of the PAMAM dendrimer. Once we analyze these shifts, we may find differences of up to 0.16 ppm, as with signal b (E-ring, methylene group). It may imply interactions caused by an external electrostatic interaction, as well as hydrogen bonding between CPT and the PAMAM dendrimer. As a result, the NMR data support effective CPT internalization within the dendrimer structure rather than a physical mixture and are in accordance with previous literature data [23,29].

Using also ^1^H NMR measurements, the number of CPT encapsulated within the dendrimer cavity was calculated. The integration ratio of the proton signals at 2.17 ppm (methylene protons near the carbonyl group; 248 H) to this at 7.43 ppm corresponding to the proton of the encapsulated CPT (pyridone moiety) determined the average number of CPT entrapped into the interior of dendrimer to be 2.4 per PAMAM molecule. The achieved result was in agreement with the calculated stoichiometry of the PAMAM dendrimer/CPT complex (Equation (6)). The slight difference could be attributed to an inaccuracy in the integration of the corresponding proton signals in the ^1^H NMR spectrum.

The hydrodynamic size of the PAMAM dendrimer/CPT complex was measured by DLS, which was found to be 140.1 ± 1.2 nm with a PDI of 0.341 ± 0.0171 (Appendix A). As was revealed in the previous literature data and confirmed by our molecular modeling investigations (see Section 2.6), the hydrodynamic size or z-average diameter of the G4.0 PAMAM dendrimer even with the encapsulated drug is around 5 nm [30]. The differences in the collected size values can be explained by the fact that DLS provides the average hydrodynamic size of particles in a solution, which is directly related to the particles’ diffusive motion. As a result, the size determined by DLS includes the solvent layers surrounding the individual nanoparticles or aggregate structures [31]. Furthermore, since the scattering intensity is directly proportional to the sixth power of the particle radius, the DLS technique is extremely sensitive to the presence of small aggregates. However, the data obtained in the presented study are still in the range of being able to cause toxicity in cancer cells because the tumor diameter cut-off is around 200–700 nm [32].

### 2.2. In Vitro CPT Release Evaluation

In vitro CPT release was conducted in phosphate-buffered saline media at two different pH values (6.50 ± 0.05—acidic conditions in the endosomes of cancer cells) and 7.40 ± 0.05—physiological conditions in the blood) to investigate the stability and interaction of the PAMAM dendrimer/CPT complex. Figure 3 and Figure 4 demonstrate the obtained data.

CPT was released in an initial value of nearly 27% for a period of 1.5 h at pH 7.40 ± 0.05, which later increased up to 70% for a duration of 8.5 h. Notwithstanding, 86% of CPT was released after 21 h and was generally maintained at this level until the end of the process, i.e., up to 168 h (Figure 3). The CPT release profile at pH 6.50 ± 0.05 did not differ significantly from the active ingredient release profile at higher pH: 68% of CPT was released after 8.5 h and nearly 76% after 21 h of incubation (Figure 4). For the time interval of 28 to 168 h, the maximum amount of CPT released was around 81%. However, it is worth noting that CPT was released in a slow and controlled manner for both conditions. In our study, we observed a slightly lower percentage of drug release at lower pH compared to physiological conditions, due to its encapsulation inside the aminofunctional PAMAM dendrimer [10]. Modifying the incubation environment to a lower pH value would presumably make the difference in CPT release profiles from the synthesized complex more noticeable. It should be stated, however, that lowering the incubation medium to a pH of 6.50 ± 0.05 was sufficient for the release of CPT in the pharmacologically active lactone form; the carboxylic form of CPT was not present under the conditions utilized (Figure 5a). Normal physiological conditions caused CPT to occur mainly in the inactive carboxylic form, which was consistent with previous literature reports (Figure 5b) [7,33]. Appendix Asummarizes the percentage content of the carboxylic form versus the lactone form at pH 7.40 ± 0.05.

To evaluate the kinetics and mechanism of CPT release from the synthesized PAMAM dendrimer/CPT complex, the obtained data were fitted to mathematical models, namely zero-order, first-order, and Korsmeyer–Peppas models (Table 1). Given the slight increase in the cumulative release during the plateau phase, kinetic calculations were performed using data from the first 8.5 h of the experiment.

The R^2^ values for the zero-order kinetics model were rather low, namely 0.7636 for pH = 6.50 ± 0.05 and 0.6863 for 7.40 ± 0.05. Based on the data in Table 1, it can be concluded that CPT was rather released at both utilized conditions with first-order kinetics, R^2^ = 0.8489 and R^2^ = 0.8958, respectively. The obtained results were in line with our expectations. Data from the literature indicate that active substances can be released with first-order kinetics from PAMAM dendrimer/drug complexes [34]. In the Korsmeyer–Peppas model, the value of n characterizes the release mechanism of the drug. For pH = 6.50 ± 0.05 and 7.40 ± 0.05, the n value corresponds to non-Fickian transport (0.45 < n = 0.89). The R^2^ values for pH = 6.50 ± 0.05 and 7.40 ± 0.05 were above 0.95.

### 2.3. Hemolytic Activity of the PAMAM Dendrimer/CTP Complex

The PAMAM dendrimer/CTP complex’s hemolytic activity was tested to determine its toxicity. The hemolysis rate of RBCs treated with the PAMAM dendrimer/CTP complex for 1 and 4 h ranged between 2 and 4%. The differences in hemolysis levels between incubation times were not statistically relevant (Figure 6). Similarly, no hemolytic activity was previously observed in the literature for CTP-loaded nanoparticles and irinotecan-loaded polymeric micelles [35,36]. Regardless of the increasing concentration of the examined system, the level of hemolysis was around 2%. In another study, CTP-loaded nanoparticles did not induce RBC hemolysis after 24 h of incubation when compared to free CTP, which caused 8% hemolysis [37]. Formulations with hemolysis values of less than 2% are considered to be non-hemolytic [38]. As a result, at ≤5 µg/mL concentrations, the PAMAM dendrimer/CTP complex was shown to be hemocompatible for potential intravenous administration.

### 2.4. Cell Viability Studies on Non-Small-Cell Lung Cancer Cell Line A549

Lung cancer is one of the leading causes of death in the world. Despite the development of numerous anticancer drugs, effectively treating lung cancer without increasing chemoresistance remains a challenge. Non-small-cell lung cancer (NSCLC) is the most common type of lung cancer, accounting for 80% of all cases [39]. The latest evidence has shown that CPT and its derivatives have broad antitumor activity against various tumor cells in vitro and in vivo. CPT and its derivatives have been shown to have anticancer activity against a range of cancer cells, including lung cancer [40]. As a result, the next stage of our investigation focused on the effects of the PAMAM dendrimer/CPT complex on the viability of non-small-cell lung cancer cells. Cells were exposed to the complex for 24 and 72 h. Cell viability was assessed using the CellTiter 96^®^ AQueous One Solution Cell Proliferation Assay (MTS).

The synthesized PAMAM dendrimer/CPT complex was shown to reduce cell viability in a concentration- and time-dependent manner. Furthermore, the complex’s activity varied between the cell lines tested. Cancer cells were more susceptible to the complex than normal fibroblasts (Figure 7). At pH 6.50 ± 0.05, the developed complex was more active against cancer cells and fibroblasts. The viability of cancer cells treated with the complex and maintained at pH 6.50 ± 0.05 for 72 h was significantly lower (IC_50_ = 1.6 ± 0.09 μg/mL) than that of fibroblasts (IC_50_ = 8.69 ± 0.68 μg/mL) (Figure 7b; Table 2). Regardless of pH, the complex was more active after 72 h (Table 2). The viability of normal fibroblasts and lung cancer cells (A549) treated for 24 h with the PAMAM dendrimer/CTP complex at pH 7.4 ± 0.05 (a) and pH 6.5 ± 0.05 (b) is shown in Appendix A, Appendix A.

The IC_50_ values of the PAMAM dendrimer/CPT complex and free CPT revealed differences in their toxicity profiles (Table 2). According to the investigation, the complex’s activity was lower than that of CPT alone, particularly in cancer cells (Figure 8). After 72 h of incubation, the difference was particularly noticeable. However, at pH 6.5 ± 0.05 after 24 h, the complex’s activity against cancer cells was higher (IC_50_ 19.07 ± 2.03 μg/mL) than the free drug (IC_50_ > 100 μg/mL) (Table 1). Normal fibroblasts showed a similar effect (Appendix A).

Other research groups discovered comparable results. Thiagarajan and coworkers found a significant difference in the toxicity profiles of free CPT and a PAMAM/CPT conjugate against colorectal cancer cells. They revealed that free CPT was 50 times more active than the PAMAM/CPT conjugate [41]. Peng et al. also observed a similar effect, reporting that the PAMAM/acetylshikonin complex had lower bioactivity in leukemia and breast cancer cells than free acetylshikonin. Nonetheless, they learned that acetylshiconin was more soluble in the complex than in the free drug [42]. Furthermore, He et al. reported that the activity of an RGD-modified PAMAM dendrimer/doxorubicine (DOX) complex was lower than that of free DOX [16].

Our findings show that the anticancer activity of the PAMAM dendrimer/CPT complex is solely due to the encapsulated CPT and that much more of the drug is needed to be released from the complex to achieve the same level of cytotoxicity as free CPT. We conclude that the free drug could enter tumor cells quickly and exert antitumor activity, suppress DNA replication, and thus induce apoptosis in tumor cell lines, whereas CPT encapsulated in the PAMAM dendrimer was internalized into cells via endocytosis in a time-dependent manner, and the drug effect was relatively slow [43]. However, as observed in the previous studies, the drug encapsulated in dendrimers is intended to protect it from inactivation while also increasing its solubility and paracellular transport. Furthermore, the process can alter the kinetics of drug entry into the cell or the release profile from the dendrimer, resulting in decreased toxicity [10,16]. It is also important to note that the release of CPT from the developed PAMAM dendrimer complex increased the availability of active CPT, whereas free CPT was found to quickly convert to the pharmacologically inactive carboxylate form, particularly at pH 7.40 ± 0.05 (see Section 2.2).

### 2.5. The Effect of PAMAM Dendrimer/CTP Complex on Cell Migration

A wound-healing assay was used to assess the effect of the PAMAM dendrimer/CPT complex on cell migration. The synthesized product was found to inhibit cancer cell migration in a concentration-dependent manner. The cells migrated into the wound area in the control group, and the wound edges became indistinguishable. Complex-treated cells, on the other hand, exhibited slower wound healing. The motility of untreated cancer cells was significantly higher than that of treated cells maintained at pH 7.40 ± 0.05 (Figure 9b). Statistical analysis revealed a significant difference in the motility of cancer cells treated with 2.5 μg/mL of the PAMAM dendrimer/CPT complex versus free CPT (Figure 9a). The motility of complex-treated A549 cells was lower than that of normal fibroblasts. However, the difference was not statistically significant. Normal fibroblasts treated with the PAMAM dendrimer/CPT complex migrated at a slower rate than untreated cells (Appendix A).

Similar findings have been reported in the literature; for example, lung cancer cells (A549) exposed to N-acetyl-D-glucosamine-labeled dendrimer (NAG-Dend)-loaded CPT (NAG-Dend-CPT) expressed slower migration of the cells into the wound area than control cells [44]. Yi and colleagues observed similar results for breast cancer cells treated with a 2-(dimethylamino)ethylmethacrylate (DEA)-modified *N*-(2-hydroxypropyl) methacrylamide (HPMA) copolymer-CPT conjugate (P-DEA-CPT); treated cells migrated at a significantly slower rate when compared to the control group [45]. The motility of prostate cancer cells (PC-3) treated with a CPT/bombesin analog was also reduced in comparison to the control group [46]. All of the studies mentioned above discovered that free CPT had a stronger inhibitory effect on cell migration than the developed products. In the current study, both free CPT and the developed complex inhibited A549 cells (Figure 9, Appendix A) and normal fibroblasts (Appendix A). These findings confirmed that the PAMAM dendrimer/CPT complex could effectively suppress lung cancer cell migration, indicating that further research is needed and promising.

### 2.6. Molecular Modeling of PAMAM Dendrimer/CPT Complexes

#### 2.6.1. Molecular Docking

Molecular modeling of the PAMAM dendrimer/drug complexes is considered challenging but also informative for exploring the conformational mobility of dendrimers and atomic specific interactions during the dendrimer–drug association. Dey and Warner [47] have determined experimentally that the lactone (neutral) and carboxylic (as a negative ion) forms are present in the molar ratios of 54:46 and 12:88 at pH 6.50 ± 0.05 and 7.40 ± 0.05, respectively, under the conditions set in their dissolution studies. As a result, both forms of CPT—lactone (L-CPT) and carboxylic (C-CPT)—were used in the molecular modeling studies. The first step in the molecular modeling chunk of this study was the molecular docking of CPT molecules in both states, L-CPT and C-CPT, to the fourth-generation PAMAM dendrimer (G4.0 PAMAM). To investigate how the complex’s stoichiometry affects the energy of formation, we docked one to five molecules of either L-CPT or C-CPT into the optimized structure of PAMAM. Table 3 and Figure 9 and Figure 10 show the results of those calculations.

The adsorption energies and optimized structures show that the form of CPT, either lactone or carboxylic, has a significant impact on the shape and stability of the complexes.

Seeing as L-CPT molecules are neutral in terms of electric charge, they are also more hydrophobic than C-CPT. As a result, they can easily penetrate the nonpolar cavities of the PAMAM dendrimer (Figure 10). The carbonyl groups and interior amides interact to encapsulate the drug, but the terminals are not involved in the interactions. Hydrogen bonds are formed between the L-CPT molecules, primarily between the carbonyl groups of the PAMAM dendrimer as H-bond acceptors and the hydroxyl groups of the L-CPT molecules as H-bond donors. Another important type of an H-bond in this complex is the one that exists between the interior amides as H-bond donors and the carbonyl groups of L-CPT as H-bond acceptors. The adsorption energies resulting from the complex formation are negative, regardless of the number of docked L-CPT molecules, indicating that such systems are stable. However, the energy per molecule drops significantly between 3 and 4. This suggests that the most stable complex is the one with the 1:3 stoichiometry of the PAMAM dendrimer/L-CPT. This value is in agreement with the experimental one (3.1 molecules of L-CPT per one molecule of PAMAM dendrimer).

The structure of the complex formed by PAMAM with C-CPT (Figure 11) differs significantly from that formed with L-CPT. Because the C-CPT are negatively charged, the main interaction is electrostatic, with the ionized carboxylic group of C-CPT interacting with the protonated terminal amine groups of PAMAM. As a result, the C-CPT molecules adsorb on the surface of the PAMAM dendrimer rather than penetrate its interior cavities. This excludes the formation of other intermolecular interactions between the remaining C-CPT atoms and the PAMAM molecule. While the absolute value of the adsorption energy for the 1:1 complex is higher for C-CPT than for L-CPT, the formation of the complex with L-CPT is more energetically favorable for each subsequent molecule than for C-CPT.

Other experimental findings of this work are explained at the molecular level by the molecular docking results. First, the energetic preference of the complex with L-CPT over the complex with C-CPT explains why the only form observed in the dissolution studies conducted at pH 6.50 ± 0.05 was the L-CPT, despite the fact that the molar ratio of L-CPT to C-CPT should be 54:46 at this pH condition, according to [44]. The simulation model structures also explain why C-CPT (at pH = 7.40 ± 0.05) releases faster than L-CPT (at pH = 6.50 ± 0.05) (i.e., 27% and 17% after 1.5 h), as C-CPT is mostly located on the surface of PAMAM and can thus dissociate from this complex more easily. It is also worth noting that the PAMAM dendrimer not only encapsulates the drug but also orientates the CPT molecules in a stabilized dispersion that disrupts cohesion and prevents clustering and subsequent recrystallization. As a consequence, the addition of PAMAM increases CPT solubility.

#### 2.6.2. Molecular Dynamics Simulations

Based on the MD simulations for the systems containing three molecules of CPT, either lactone (L-CPT) or carboxylic (C-CPT) form, some geometric observables for the complexes were determined, such as the mean gyration radius (R_g_), solvent-accessible surface area (SASA), and radial distribution functions G(r).

The R_g_ can be used to characterize the size and degree of curling of polymer chains. Opitz and Wagner used molecular dynamics simulations to investigate the molecular structure of PAMAM dendrimers and discovered that the gyration radius measured by the experiment was in good agreement with the simulation results [48]. The experimentally determined R_g_ of the PAMAM dendrimer G4.0 at neutral pH was 22.5 [49], while the R_g_ for L-CPT and C-CPT PAMAM complexes in this study were 21.4 and 22.6, respectively. The decrease in R_g_ during complex formation with L-CPT is caused by intermolecular forces between the host and guest molecules located in the cavities of the PAMAM dendrimer. The slight expansion of the PAMAM dendrimer upon complexation with C-CPT results from the presence of C-CPT molecules adsorbed at its surface.

To calculate the SASA, a probe radius of 1.4 was used as the van der Waals radius of a water molecule. The results for complexes with L-CPT and C-CPT were 14504.5 Å2 and 14705.9 Å2, respectively. These results are comparable to those obtained for the PAMAM dendrimer G4.0 at pH = 7.40 ± 0.05, namely 14862.4 Å2 [50]. The results for complexes with L-CPT and C-CPT are similar because pH is a major factor that influences the SASA of PAMAM molecules, and the two studied systems were prepared to correlate with the results obtained at neutral pH, with the charged terminal amine groups.

The radial distribution function, g(r), is calculated by averaging the distribution of interatomic vector lengths on a sphere of radius (r) and thickness (Δr). We used (Δr) = 0.1 Å, with the radius of the sphere, r, varying between 1 and 23 Å. Figure 12 depicts the radial distribution functions, g(r), of PAMAM dendrimer complexes with L-CPT and C-CPT. A high-density core region appears near the center, while the terminal groups of the PAMAM dendrimer are distributed throughout the molecule. Furthermore, this analysis confirms the presence of backfolded branches in the dendritic structure [51].

## 3. Materials and Methods

### 3.1. Materials

(*S*)-(+)-Camptothecin ((*S*)-4-Ethyl-4-hydroxy-1H-pyrano-[3′,4′:6,7]indolizino[1,2-b]quinoline-3,14(4H,12H)-dione (CPT) (purity: >97.0% (HPLC)) was purchased from TCI Europe N.V. (Zwijndrecht, Belgium). PAMAM dendrimer, ethylenediamine core, generation 4.0, 10 wt.% solution in methanol was obtained from Sigma-Aldrich, Co., Poznan, Poland. *N*,*N*-Dimethylformamide (DMF, anhydrous, 99%, Avantor Performance Materials S.A., Gliwice, Poland), dimethyl sulphoxide (DMSO, anhydrous, 99%, Avantor Performance Materials S.A., Gliwice, Poland), and acetonitrile ((ACN), anhydrous, 99.8%, Avantor Performance Materials S.A., Gliwice, Poland) were used as received. Dimethyl-*d*_6_-sulfoxide (DMSO-*d*_6_) in ampoules, for NMR measurements (99.9 atom% D), was purchased from ARMAR Chemicals (Döttingen, Switzerland). Phosphate buffer solutions (pH 7.40 ± 0.05 and pH 6.50 ± 0.05, 20 °C, Avantor Performance Materials S.A., Gliwice, Poland) were also used as received.

### 3.2. PAMAM Dendrimer/CPT Complex Synthesis

A total of 9.1 mg of CPT was dissolved in 2.5 mL of DMF after stirring for 10 min. One milliliter of PAMAM dendrimer (10 wt.% solution in methanol) was then added to this solution. The flask was incubated in dark for 24 h at 37 °C and shaken at 100 rpm. Following that, DMF was evaporated, and 2 mL of distilled water was added to the dry residue and stirred at room temperature for 1 h to precipitate uncomplexed CPT. The unreacted CPT was filtered through a syringe filter, and the residual water was evaporated over 2 h using rotary evaporation. The resulting compound was dried in a vacuum oven for 24 h until it reached constant weight. The yield of the synthesized complex was 73%. The PAMAM dendrimer/CPT complex was stored in the dark until further usage. All experiments were completed within 24 h of preparing the material.

Data for PAMAM dendrimer/CPT: ^1^H NMR (DMSO-*d*_6_, 700 MHz; δH, ppm): 8.61 (**g** of CPT), 8.15 (**k** of CPT), 8.05 (**h** of CPT), 7.91 (broad, **8** of PAMAM dendrimer), 7.68 (**j** of CPT), 7.65 (**i** of CPT), 7.46 (**a** of CPT), 5.26 (**e** of CPT), 5.17 (**f** of CPT), 3.10–3.02 (broad, **3** + **5** of PAMAM dendrimer), 2.62 (**1** of PAMAM dendrimer), 2.54 (**6** + **7** of PAMAM dendrimer), 2.40 (**4** of PAMAM dendrimer), 2.17 (**2** of PAMAM dendrimer), 2.01 (**b** of CPT), 0.83 (**c** of CPT).

### 3.3. Drug Content in the PAMAM Dendrimer/CPT Complex

HPLC measurements (see below) were used to determine the quantity of CPT entrapped in the complex. In brief, excess of acetonitrile was added to the PAMAM dendrimer/CPT complex, and it was then placed in an ultrasonic bath for 30 min. The material was then filtered with a syringe filter before being evaluated with HPLC. The *EE* was estimated as follows (1):*EE*(%) = (Q_encapsulated_/Q_initial_) × 100(1)
where Q_initial_ is the initial amount of CPT added, and Q_encapsulated_ is the amount of entrapped CPT in the complex [52].

### 3.4. CPT Release Study from the PAMAM Dendrimer/CPT Complex

The synthesized and characterized dendrimer PAMAM/CPT complex was suspended in 2.0 mL of phosphate buffer at pH 6.50 ± 0.05 and 7.40 ± 0.05 resulting in initial concentrations of 17.41 and 18.08 mg/mL. The obtained materials were poured into dialysis membranes with a pore diameter of 3.50 kD and immersed in the appropriate volume of buffer solutions. The release procedure was carried out at 37 °C with continuous rotation at 100 (cycles/min). At 1.5; 3; 6; 8.5; 21; 28; 48; 75; 98; and 168 h of the process, 3.0 mL samples were taken and supplemented with a new portion of fresh buffer of appropriate pH. The amount of drug released, as well as the percentages of carboxylic and lactone CPT, was determined using HPLC at 363 nm and gradient elution. Calibration curve was designed between 0.46 and 46.55 µg/mL (R^2^ = 0.9993; y = 0.8379x + 0.3333) (Appendix A). The samples were placed into auto-sampler vials, capped, and placed in the HPLC auto-sampler. Each sample was tested three times, and results were expressed as the amount of CPT released (in μg).

### 3.5. Mathematical Models

The kinetics and mechanism of CPT release were determined by fitting experimental data to theoretical mathematical models—zero-order, first-order, and Korsmeyer–Peppas—using the following equations:

Zero-order model:F = kt(2)

First-order model:logF = log F_0_ − kt/2.303(3)

Korsmeyer–Peppas model:F = kt^n^ (F < 0.6)(4)
where:

F—the fraction of CPT released up to time (t);

F_0_—the initial concentration of CIP;

k—the constant of the mathematical models;

n—the exponent of the Korsmeyer–Peppas model [53,54].

### 3.6. Hemolysis Assay

Blood samples were collected from healthy volunteers and placed in K2-EDTA-coated tubes to avoid coagulation. The blood was centrifuged for 10 min at 2500 rpm. The hematocrit and plasma levels were recorded on the tube. The plasma was gently aspirated, and PBS (pH 7.40 ± 0.05) was added up to the marked level of plasma. The solution was gently mixed and centrifuged for 10 min at 2500 rpm. The washing procedure was repeated twice. After the last washing, the supernatant was aspirated and mixed with fresh PBS (pH 7.40 ± 0.05). RBCs were diluted in PBS (pH 7.4 ± 0.05) to achieve a 10% RBC suspension. To make 2% RBC suspension, 10% RBC suspension was used. A 2% RBC suspension was incubated in a 1:1 ratio with the PAMAM dendrimer/CPT complex (0.5–100 μg/mL) at 37 °C for 1 and 4 h. Samples were centrifuged at 4500 rpm for 5 min at each time point, and 100 μL of supernatant from each sample was transferred to a 96-well plate. Optical density (OD) was measured at a wavelength of 540 nm. A value of 100% hemolysis was determined by incubation of 10% in distilled water (ratio 1:9). For the negative control (0% hemolysis), a 2% RBC suspension was incubated in PBS (pH 7.4 ± 0.05) (ratio 1:1). The experiment was carried out twice. The value of peptidomimetic-induced hemolysis was calculated using the following equation:Hemolysis [%] = (A − A0%)/(A100% − A0%)·100%(5)
where A is the absorbance of the sample, A100% is the absorbance of the positive control (100% hemolysis), and A0% is the absorbance of the negative control (0% hemolysis).

The hemolysis assay was carried out with the approval of the Bioethics Committee—Commission for the Supervision of Research on People and Animals at CSK MSWiA in Warsaw (no. 67/2017).

### 3.7. Cell Culture

Non-small-cell lung cancer cell line A549 (ATCC CCL-185) and normal human dermal fibroblasts (Promocell) were cultured in Dulbecco’s Modified Eagle Medium (DMEM) (Biowest, Nuaillé, France) supplemented with 10% fetal bovine serum (FBS) (Thermo Scientific, Waltham, MA, USA) and 1% penicillin-streptomycin solution (Biowest, Nuaillé, France). Cells were maintained at 37 °C in humidifying conditions with 5% CO_2_.

### 3.8. Cell Viability Assay

The cellular metabolic activity was evaluated using the CellTiter 96^®^ AQueous One Solution Cell Proliferation Assay (MTS) (Promega, Walldorf, Germany). Cells were seeded at a density of 4·103 per well and incubated overnight. The growth media were changed the next day to DMEM at pH 7.40 ± 0.05 or DMEM at pH 6.50 ± 0.05. The cells were then treated with PAMAM dendrimer/CPT complex (0–100 µg/mL). Plates were incubated at 37 °C for 24 and 72 h. At each time point, 20 µL of MTS solution was added to each well, and the plates were incubated for 2 h at 37 °C. The absorbance was measured at ƛ = 490 nm. Untreated cells were used as a negative control, and their cell viability was defined as 100%.

### 3.9. Cell Migration (Scratch Assay)

Cells were seeded in 24-well plates at a density of 8·10^4^ cells per well and incubated for 48 h. Following that, clear lines were formed with a sterile 10 μL pipette tip. The medium was carefully removed from the cells, and fresh medium with pH 7.40 ± 0.05 or 6.50 ± 0.05 was added. PAMAM dendrimer/CTP complex or free CTP were then added. Microscopic pictures of phase contrast were taken immediately after wounding (T_0_) and after 24 h (T_24_). The cell migration rate was determined by measuring the wound area at each time point. ImageJ software was used to perform the measurement results (National Institute of Health, V1.5, Bethesda, MA, USA).

### 3.10. Statistical Analysis

For in vitro experiments, the results were presented as the mean and standard deviation of the mean. Statistical significance of the differences between means was evaluated by testing normality of distribution followed by one-way ANOVA or non-parametric Kruskal–Wallis test. To compare different time points or conditions, two-way ANOVA was followed by Bonferroni’s multiple comparisons. *p* < 0.05 was considered statistically significant. The results were analyzed using GraphPad Prism 5.0 (GraphPad Software, San Diego, CA, USA).

### 3.11. Molecular Modeling of PAMAM Dendrimer/CPT Complex

The structure of PAMAM dendrimer generation 4.0 was built using the Build Polymers module within the Biovia Materials Studio 2020 software suite (https://www.3ds.com/products-services/biovia/products/molecular-modeling-simulation/biovia-materials-studio/, accessed at 6 August 2022), the structures of lactone and carboxylic forms of CPT were built using the Visualizer Tool of Materials Studio. It has been proven that at pH values ranging from 6.5 to 9.0, the terminal, primary amino groups of the dendrimers are protonated and thus possess positive charge [55]. Therefore, all the terminal amino groups of the modeled PAMAM structure were protonated, to enable the comparison with the dissolution studies performed within this work, which were conducted at pH 6.50 ± 0.05 and 7.40 ± 0.05.

The subsequent optimization of the structures was performed using the Forcite Plus module, applying ultra-fine quality settings of geometry optimization and smart algorithm. The convergence tolerance values were set to 2 × 10^−5^ kcal/mol for energy, 1 × 10^−3^ kcal/(Å × mol) for force, and 1 × 10^−5^ Å for displacement, with 5 × 10^4^ maximum iterations. The COMPASS II force field [56] was chosen for all the calculations as it has been proven to provide accurate results while modeling the PAMAM dendrimers [57]. The charges were assigned using the COMPASS II force field as charge assignments are part of the parameters stored in this force field definition. The electrostatic and van der Waals summation methods were both atom-based with cubic spline truncation of non-bond energy terms, 18.5 Å cut-off distance, 1 Å spline width, and 0.5 Å buffer width.

Molecular docking calculations were performed using the Adsorption Locator module within the Biovia Materials Studio 2020 software suite. The Adsorption Locator module identifies possible adsorption sites by carrying out Monte Carlo searches of the configurational space of the substrate (PAMAM)–adsorbate (CPT) system as the temperature is slowly decreased within the simulated annealing process of a molecular dynamics run. The benefit of such an approach is that it enables simultaneous docking of the specified numbers of CPT, in this work ranging from 1 to 5. The Adsorption Locator utilizes a simulated annealing simulation with geometry optimizations in between successive heat–cool cycles. A total of 10 cycles containing 100 000 steps per cycle with annealing temperatures of 100 to 5000 K were used for reproducible results. The Monte Carlo parameters were set to a probability of 0.32 (ratio = 1) for ‘conformer’, ‘rotate’, and ‘translate’ while ‘regrow’ was set to 0.1 (ratio = 0.03). The COMPASS II force field was used in the calculations, with the parameters the same as those for geometry optimization, described in detail in the previous paragraph.

To calculate the stoichiometry of the PAMAM dendrimer/CPT complex, the following formula was used:(6)$$L=\frac{m_{CPT}\times M_{PAMAM}\times EE}{M_{CPT}\times V_{PAMAM}\times d_{PAMAM}\times C_{PAMAM}}$$
where L is the loading (average number of CPT molecules per one PAMAM molecule), m_CPT_ is the mass of added CPT (9.1·10^−3^ g), *M*_PAMAM_ is the molar mass of PAMAM (14.214,17 g/mol), *EE* is encapsulation efficiency (68.24%), *M*_CPT_ is the molar mass of CPT (348.35 g/mol), V_PAMAM_ is the volume of added PAMAM methanol solution (1·10^−3^ L), d_PAMAM_ is the density of the PAMAM methanol solution (0.813 g/mL), and C_PAMAM_ is the percentage concentration of PAMAM methanol solution (10%).

The optimized systems containing three molecules of CPT (either lactone or carboxylic form) were then subjected to molecular dynamics (MD) calculations under periodic boundary conditions. First, the models were solvated with explicit water molecules in a cubic simulation cell with dimensions sufficient to provide a solvation layer of at least 10 Å around the dendrimer molecule; the size of the simulation box for all systems was several times larger compared to the radius of gyration of the dendrimer. To maintain the charge neutrality, an appropriate number of H_2_PO_4_^−^ counterions was added, as the dissolution studies were conducted in the phosphate buffer. Subsequently, a simulated annealing process was undertaken in 5 cycles of 10,000 steps, within a temperature range of 300–500 K, using the NVT ensemble mode, 5 heating ramps per cycle, 100 dynamics steps per run, and the Berendsen thermostat with 0.1 ps decay constant. Additional relaxation was performed by short NPT MD which lasted 25 ps. The density and cell dimensions were monitored up to constant values that then indicated the final state of relaxation. A longer (50 ns) production run, in NVT mode, with the Nosé–Hoover thermostat for temperature control with a Q-ratio of 2 was performed with a 1 fs time step. For the MD calculations, the COMPASS II force field was used with the parameters the same as those used for geometry optimization.

### 3.12. Measurements

^1^H NMR measurements were performed using a 700 MHz Agilent DirectDrive2 spectrometer (Santa Clara, CA, USA) equipped with a room-temperature HCN probe, temperature-controlled at 25 °C. For water signal suppression, we used a PRESAT pulse sequence with the number of transients = 16, interscan delay = 2 s, 32 K complex points, acquisition time = 2.94 s, saturation delay = 2 s, and saturation power = −15.5 dB. All samples were dissolved in DMSO-*d*_6_.

The amount of CPT released in vitro was estimated using high-performance liquid chromatography (HPLC) (Beckman Coulter, Miami, FL, USA) based on previous literature report with a small modification [58]. The HPLC apparatus was equipped with an autosampler (Triathlon 900, Spark Holland B.V., Emmen, Netherlands), pomp (Beckman Coulter System Gold^®^ 125NM Solvent Module, Fullerton, CA, USA), and UV/VIS detector (Beckman Coulter System Gold^®^ 166, Fullerton, CA, USA). The analysis was performed at 363 nm with a C18 column (Luna 25 cm, 5 µm, 100 A, Phenomenex, Basel, Switzerland) using a gradient mobile phase of acetonitrile (ACN):phosphate buffer solution (pH 6.50 ± 0.05) (*v*/*v*), delivered at a flow rate of 1.0 mL/minute (ACN concentration varied with time: 5% ACN after 0 min, 15% ACN after 5 min, 35% ACN after 15 min, 50% ACN after 20 min, and 5% ACN after 22 and 25 min). The column was placed at 30 °C, and the injection volume was 20 μL. The retention time of CPT was 13.90 ± 0.1 min for carboxylic form of CPT and 20.69 ± 0.1 min for lactone form.

A dynamic light scattering (DLS) method was employed to determine the hydrodynamic size and polydispersity index (PDI) of the PAMAM dendrimer/CPT complex using a Zetasizer Nano ZS instrument (Malvern Instruments, Westborough, MA, USA) equipped with a red laser at a wavelength of 633 nm and scattering angle of 173° at 25 °C.

## 4. Conclusions

Cancer treatment challenges have resulted in ongoing efforts to develop new drug delivery systems (DDSs), also with the current investigation concentrating on the preparation of the PAMAM dendrimer/CPT complex for non-small-cell lung cancer cell targeting. This complex was synthesized and thoroughly characterized using NMR spectroscopy, which supports effective CPT internalization within the dendrimer structure. Furthermore, the integration ratio of the characteristic proton signals of the PAMAM dendrimer and CPT allowed us to calculate the average number of drug molecules entrapped inside the dendrimer to be 2.4. In vitro CPT release evaluation in physiological and acidic (cancer cell endosomes) conditions revealed that the drug was released in a slow and controlled manner for both environments. Furthermore, lowering the pH of the medium to 6.50 ± 0.05 was sufficient to release CPT only in its pharmacologically active lactone form. Fitting the obtained release data to mathematical models demonstrated that CPT was released with rather first-order kinetics and non-Fickian transport at both pH levels. The toxicity of the PAMAM dendrimer/CTP complex was determined by evaluating its hemolytic activity. The findings showed that the synthesized product is hemocompatible for potential intravenous administration at concentrations ≤ 5 µg/mL. The effect of the complex on the viability of non-small-cell lung cancer cells (A549) was then assessed, displaying that it reduces cell viability in a concentration and time-dependent manner and that cancer cells were more susceptible to the complex than normal fibroblasts. We continually investigated the effect of the synthesized product on cell migration and observed that both, free CPT and the developed complex, inhibited A549 cells and normal fibroblasts, confirming that the complex could effectively suppress lung cancer cell migration. Molecular modeling of the developed material, on the other hand, enabled us to investigate how the complex’s stoichiometry influences the energy of formation. It was calculated that the lactone or carboxylic forms of CPT had a significant impact on the shape and stability of the complexes and that the formation of the complex with the lactone form of CPT was more energetically favorable for each subsequent molecule than the carboxylic form. The obtained modeled structures also showed that the carboxylic form of CPT was mostly found on the surface of the PAMAM dendrimer and could thus dissociate from this complex more easily. Our promising findings could open the door to future research into the development of drug–dendrimer complexes designed for a specific cancer and/or organ system.

## Figures and Tables

**Figure 1 molecules-28-02696-f001:**
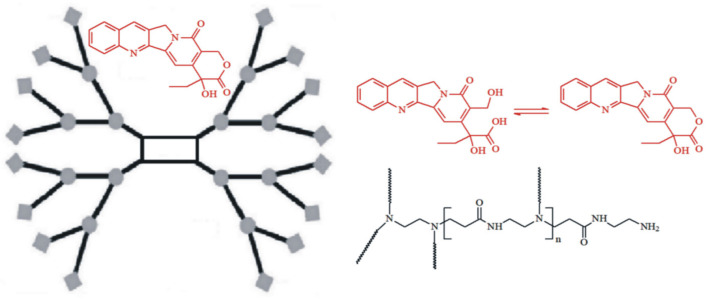
Schematic illustration of the PAMAM dendrimer G4.0 with encapsulated CPT.

**Figure 2 molecules-28-02696-f002:**
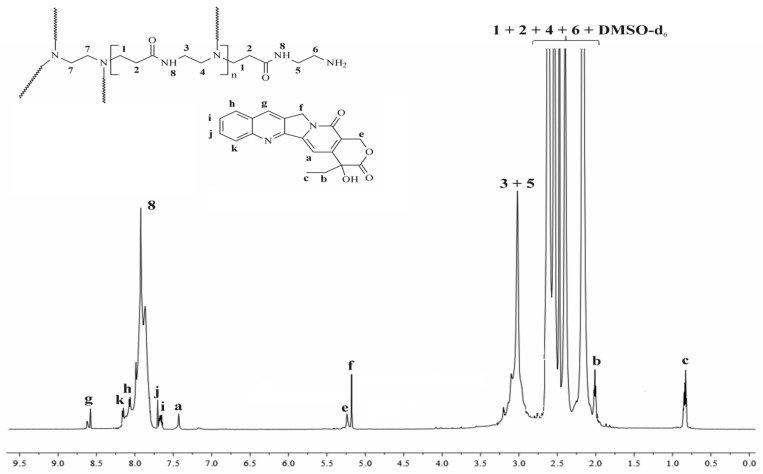
^1^H NMR spectrum of the PAMAM dendrimer/CPT complex.

**Figure 3 molecules-28-02696-f003:**
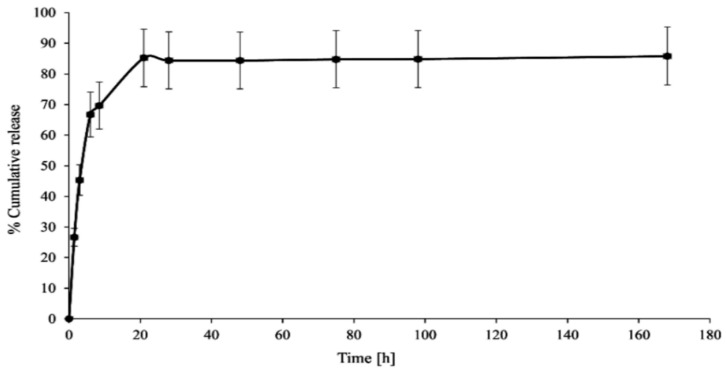
In vitro CPT release profile from the synthesized complex (pH 7.40 ± 0.05).

**Figure 4 molecules-28-02696-f004:**
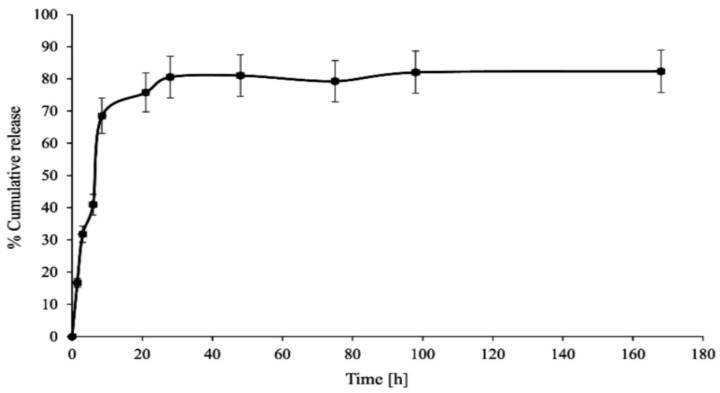
In vitro CPT release profile from the synthesized complex (pH 6.50 ± 0.05).

**Figure 5 molecules-28-02696-f005:**
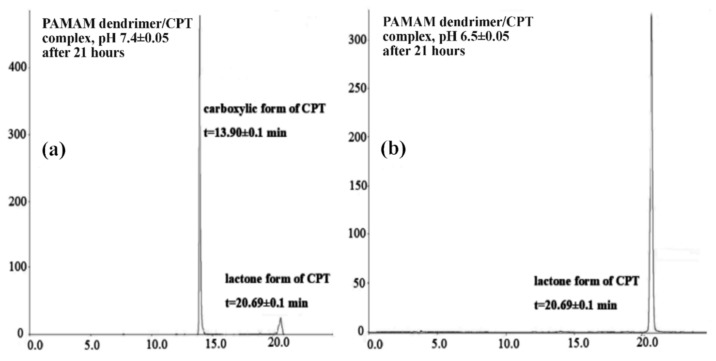
HPLC chromatograms of the CPT released in the lactone and carboxylate forms (**a**) pH = 7.4; (**b**) pH = 6.5.

**Figure 6 molecules-28-02696-f006:**
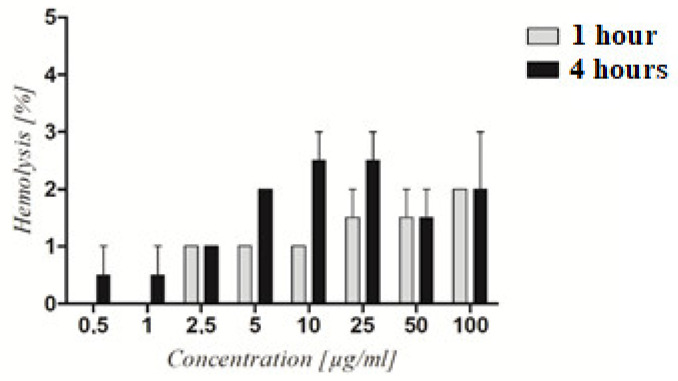
Hemolytic activity of the PAMAM dendrimer/CTP complex. The graph depicts the level of hemolysis of RBC treated with increasing concentrations of the complex after 1 and 4 h of treatment. Statistical analysis using two-way ANOVA followed by the Bonferroni post-test revealed no significant differences (*p* > 0.05) between the obtained results.

**Figure 7 molecules-28-02696-f007:**
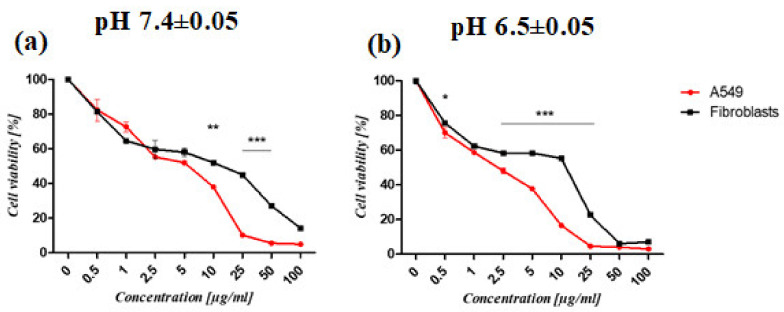
The viability of normal fibroblasts and lung cancer cells (A549) treated for 72 h with PAMAM dendrimer/CTP complex at pH 7.4 ± 0.05 (**a**) and pH 6.5 ± 0.05 (**b**). The graphs depict differences in the susceptibility of cells to the complex. The MTS assay was used to determine the relative cell number. The results are given as mean ± SEM. Two-way ANOVA was used for statistical analysis, followed by Bonferroni post-tests. When the following conditions were met, the results were considered statistically significant: * *p* < 0.05; ** *p* < 0.01; *** *p* < 0.005.

**Figure 8 molecules-28-02696-f008:**
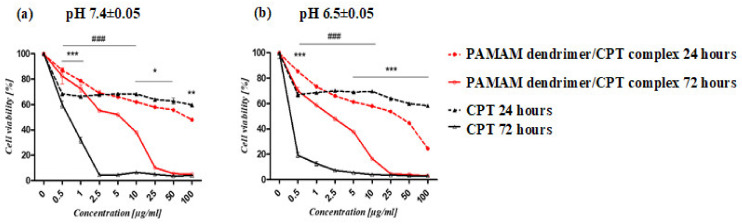
The viability of lung cancer cells (A549) treated with PAMAM dendrimer/CPT complex and CPT at pH 7.4 ± 0.05 (**a**) and pH 6.5 ± 0.05 (**b**). The graphs show differences in cell susceptibility to the complex and CPT after 24 (*) and 72 (#) hours of treatment. The MTS assay was used to calculate the relative cell number. The results are presented as mean ± SEM. The statistical analysis was carried out using two-way ANOVA, followed by Bonferroni post-tests. When the following conditions were met, the results were considered statistically significant: * *p* < 0.05; ** *p* < 0.01; *** *p* < 0.005.

**Figure 9 molecules-28-02696-f009:**
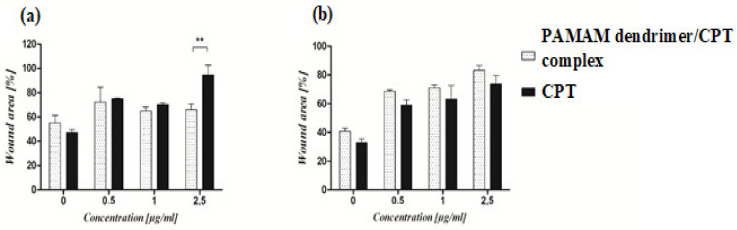
The effect of PAMAM dendrimer/CPT complex and free CPT on the migration of cancer cells maintained at pH 7.40 ± 0.05 (**a**) and 6.5 ± 0.05 (**b**). The graphs show the migration rate of cells treated with the complex or CPT for 24 h. For each concentration, the percentage of wound area was calculated in comparison to the control wound (T0). The results are presented as mean ± SEM. ** *p* < 0.01.

**Figure 10 molecules-28-02696-f010:**
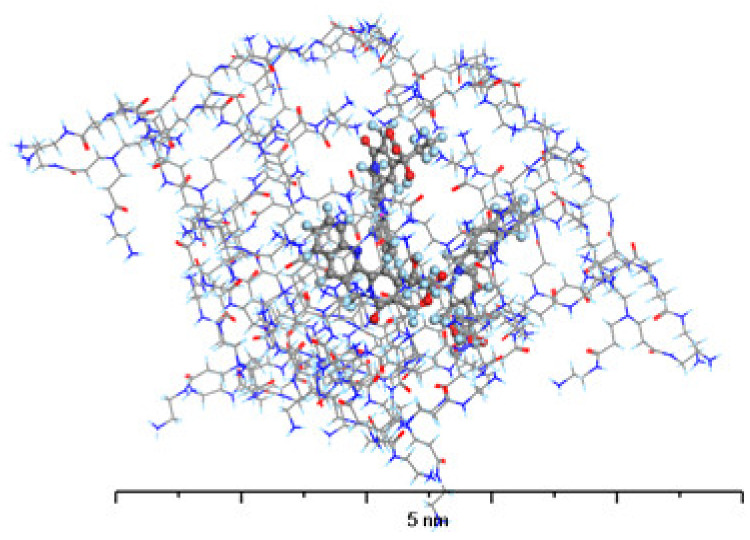
The structure of G4-PAMAM loaded with 3 molecules of L-CPT. Atom coloring: Hydrogen—light blue, Nitrogen—dark blue, Carbon—gray, Oxygen—red.

**Figure 11 molecules-28-02696-f011:**
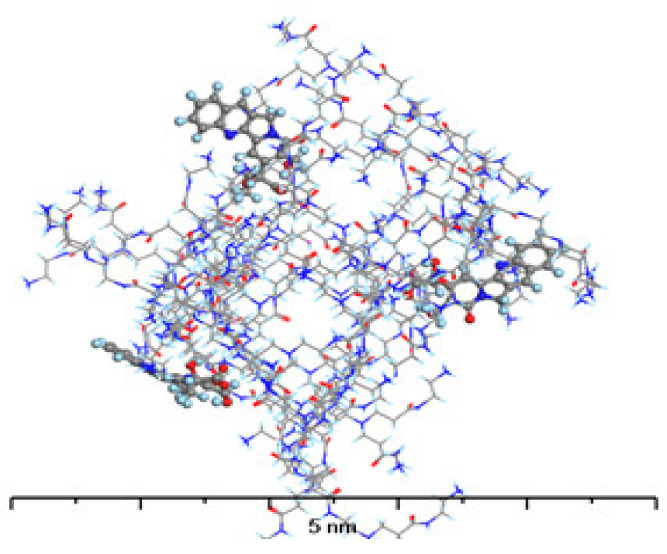
The structure of G4-PAMAM loaded with 3 molecules of C-CPT. Atom coloring: Hydrogen—light blue, Nitrogen—dark blue, Carbon—gray, Oxygen—red.

**Figure 12 molecules-28-02696-f012:**
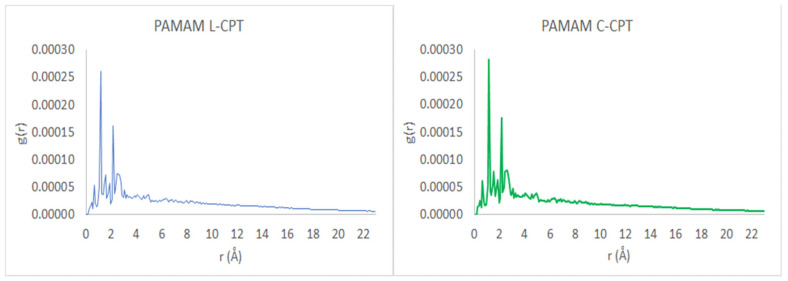
Radial distribution function, g(r), as a function of the radius, r, of the spherical shell over which the interatomic vector lengths are averaged.

**Table 1 molecules-28-02696-t001:** Data analysis of CPT release from PAMAM dendrimer/CPT complex in different pH values.

	Zero-Order Model	First-Order Model	Korsmeyer–Peppas Model		Drug Transport Mechanism
	*R* ^2^	*R* ^2^	*R* ^2^	n	
PAMAM dendrimer/CPT complex (pH = 6.50 ± 0.05)	0.7636	**0.8489**	**0.9557**	0.747	non-Fickian transport
PAMAM dendrimer/CPT complex (pH = 7.40 ± 0.05)	0.6863	**0.8958**	**0.9637**	0.569	non-Fickian transport

**Table 2 molecules-28-02696-t002:** Data of the IC_50_ (µg/mL) of the PAMAM dendrimer/CPT complex.

		pH = 7.40 ± 0.05		pH = 6.50 ± 0.05	
		24 h	72 h	24 h	72 h
Lung cancer cells (A549)	PAMAM dendrimer/CPT complex	>100	4.71 ± 0.32	19.07 ± 2.03	1.6 ± 0.09
	CPT	>100	0.61 ± 0.01	>100	0.16 ± 0.02
Normal fibroblasts	PAMAM dendrimer/CPT complex	>100	15.2 ± 3.17	13.89 ± 1.08	8.69 ± 0.68
	CPT	>100	2.17 ± 0.12	>100	1.48 ± 0.06

**Table 3 molecules-28-02696-t003:** The results of the molecular docking of CPT in both states, lactone (L-CPT) and carboxylic (C-CPT), on PAMAM dendrimer generation four (G4.0).

	L-CPT		C-CPT	
Number of CPT Molecules	Total Adsorption Energy (kcal/mol)	Energy per Molecule of L-CPT (kcal/mol)	Total Adsorption Energy (kcal/mol)	Energy per Molecule of C-CPT (kcal/mol)
1	−124.67	−124.67	−145.13	−145.13
2	−237.66	−118.83	−149.06	−74.53
3	−348.75	−116.25	−180.57	−60.19
4	−336.80	−84.20	−169.80	−42.45
5	−225.65	−45.13	−142.50	−28.50

## Data Availability

The data presented in this study are available on request from the corresponding author.

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
