# Peer review of "Poly(amidoamine) Dendrimer/Camptothecin Complex: From Synthesis to In Vitro Cancer Cell Line Studies"

_molecules, 2023, doi:10.3390/molecules28062696_

Round 1

Reviewer 1 Report

The manuscript molecules-2230181 "Poly(amidoamine) dendrimer/camptothecin complex: From synthesis to in vitro cancer cell line studies" by Oledzka et al. describes the study of complexation of camptothecin (CPT) with 4th generation of PAMAM dendrimer and in vitro antitumor activity of the obtained complex. The obtained results have been confirmed by NMR spectroscopy and DLS. Despite the fact that the complexation of CPT by PAMAM dendrimers has been studied earlier, this manuscript may be of interest to readers of Molecules.

Reviewer’s comments:

1) The abstract is large and more like conclusions. I recommend to reduce the abstract for a better understanding of readers.

2) The authors write that many examples of the creation of PAMAM-CPT systems are available in the literature. Please write more clearly how these results differ from those previously obtained. What is the main novelty of this manuscript?

3) The chemical shifts of protons are not visible in Figure 2. I recommend that the authors draw the NMR spectra of the PAMAM dendrimer, camptothecin and their mixtures in one figure. Shift the corresponding signals depict graphically (lines, areas, etc.).

4) Two-dimensional NMR (e.g., 1H-1H NOESY) spectroscopy is often used to establish the complexes’ structure. Why didn't the authors use it?

5) I recommend the authors to strengthen the Introduction part about the synthesis of PAMAM dendrimers derivatives and their complexation properties. Recent articles on this topic should be added, e.g., 10.3390/ijms24010819, 10.3390/pharmaceutics14122748, 10.1016/j.onano.2022.100053.

6) Minor changes:

- "in vitro"/"in vivo" should be italic "in vitro"/"in vivo".

- "N,N-Dimethylformamide" should be "N,N-Dimethylformamide".

- "DMSO-d6" should be "DMSO-d6".

- Please double check errors and typos, especially, upper and lower case.

Author Response

Kindly please find attached our responses to your criticisms.

Reviewer 2 Report

The presented manuscript by Oledzka and colleagues shows an investigation of the well-known anticancer alkaloid camptothecin (CPT) in conjugate with PAMAM dendrimer. The authors provided a wide raged studies starting from the synthesis of the CPT-PAMAM complex, its characteristics (size, structure, stoichiometry, release profile), and finishing with the biological investigation (viability and motility assays, hemolytic activity). The manuscript is very complex but clearly and logically planned. In my opinion, it is suitable for publication in Molecules after correction.

1.       I strongly suggest to the authors re-write some chapters of the manuscript. I feel like I've read the same thing several times. For example, I suggest shortening the last part of the Introduction. In my opinion, there is a place to the encouragement of the reader to read the paper. This part of the Introduction is too detailed in the present form. I also suggest rethinking and rewriting of Conclusions chapter.

2.       There are a lot of misspellings within the text.

Author Response

(The authors gave the same response as above.)

Reviewer 3 Report

The present manuscript dealing with the encapsulation of a CPT into a PAMAM dendrimer is a standard and solid study. I recommend the work for publication after minor revisions and questions answered.

- Is there a value in the literature for the solubility of CPT in water (buffer) or is it completely insoluble?
- For better clarity, the graphs in Figure 4 should have cut-outs showing the release of CPT max. up to the 30th minute.
- Line 291: ..... "fibroblasts (IC50 = 8.69±0.68 μg/ml) (Figure 2 b; Table 2)....." . figure 2 b does not exist.

- Line 503: ..."The yield of the synthesized complex was 73 %."... to what was the yield related? To initial amound of PAMAM?

Author Response

(The authors gave the same response as above.)

Round 2

Reviewer 1 Report

I thank the authors for answering all my questions.

I also recommend authors to double-check the manuscript for typos and errors during proofreading.